# Crosstalk between Thyroid Hormone and Corticosteroid Signaling Targets Cell Proliferation in *Xenopus tropicalis* Tadpole Liver

**DOI:** 10.3390/ijms232213715

**Published:** 2022-11-08

**Authors:** Muriel Rigolet, Nicolas Buisine, Marylou Scharwatt, Evelyne Duvernois-Berthet, Daniel R. Buchholz, Laurent M. Sachs

**Affiliations:** 1UMR PhyMA CNRS, Muséum National d’Histoire Naturelle, 75005 Paris, France; 2Department of Biological Sciences, University of Cincinnati, Cincinnati, OH 45221, USA; 3UMR7221 CNRS, Muséum National d’Histoire Naturelle, CP32, 7 Rue Cuvier, CEDEX 05, 75231 Paris, France

**Keywords:** thyroid hormone, glucocorticoid, hormonal crosstalk, metamorphosis, *Xenopus tropicalis*

## Abstract

Thyroid hormones (TH) and glucocorticoids (GC) are involved in numerous developmental and physiological processes. The effects of individual hormones are well documented, but little is known about the joint actions of the two hormones. To decipher the crosstalk between these two hormonal pathways, we conducted a transcriptional analysis of genes regulated by TH, GC, or both hormones together in liver of *Xenopus tropicalis* tadpoles using RNA-Seq. Among the differentially expressed genes (DE), 70.5% were regulated by TH only, 0.87% by GC only, and 15% by crosstalk between the two hormones. Gene ontology analysis of the crosstalk-regulated genes identified terms referring to DNA replication, DNA repair, and cell-cycle regulation. Biological network analysis identified groups of genes targeted by the hormonal crosstalk and corroborated the gene ontology analysis. Specifically, we found two groups of functionally linked genes (chains) mainly composed of crosstalk-regulated hubs (highly interactive genes), and a large subnetwork centred around the crosstalk-regulated genes *psmb6* and *cdc7*. Most of the genes in the chains are involved in cell-cycle regulation, as are *psmb6* and *cdc7*, which regulate the G2/M transition. Thus, the biological action of these two hormonal pathways acting together in the liver targets cell-cycle regulation.

## 1. Introduction

The endocrine system acts to integrate and coordinate different specialized cell groups or organs in multicellular organisms. Being part of this system, thyroid hormones (THs) and glucocorticoids (GCs) control metabolism and regulation of the internal environment [1,2]. They also mediate the growth and maturation of the organism, as well as the responses to the environment. Their respective actions are commonly subject to interactions because the same cell can often respond to more than one hormone. These interactions are linked to stress response and are often overlooked. However, they can also lead to tradeoffs and can even disrupt development of the structures and functions of the organism [3]. The biological processes specifically targeted by functional interactions between the two signaling pathways are poorly known. This paper aims to shed light on this gap. We hypothesize that this knowledge concerning these interactions will allow us to distinguish between beneficial and adverse effects of altered hormone signaling.

THs and GCs act mainly via transcriptional regulation of gene expression. They bind to the two TH receptors (TR alpha and beta) and the two GC receptors (GR and MR) that are transcription factors belonging to the super family of nuclear receptors [4]. Many studies have addressed the effects and mechanisms of action of TH and GC used individually [5,6]. This reductionism proved to be extremely powerful. THs control the expression of several genes involved in development, metabolism, and growth of most organs throughout the life cycle [7]. GCs mediate physiological processes including immune responses, stress responses, metabolism, and electrolyte homeostasis [8]. Both hormones regulate cellular processes such as mitosis, cell differentiation, and apoptosis. They also target common organs, and both modulate their functions. Thus, functional interactions likely exist between these two endocrine pathways, although data on such TH and GC interactions are scarce. An established interaction is that GCs can act on TH availability through the regulation of TH metabolizing enzymes [9,10]. Furthermore, THs and GCs can directly regulate the expression of common target genes such as the *klf9* gene, through a “synergy module” including both hormone response elements [11].

Development of Xenopus liver during metamorphosis is an ideal vertebrate model system for dissecting TH and GC interactions. First, mechanisms of TH and GC action are conserved in vertebrates. Studies in Xenopus have greatly contributed to knowledge on the mechanisms behind TH function [12,13]. Second, anuran amphibian metamorphosis is a post-embryonic developmental period when a number of morphological, developmental, biochemical, and physiological processes affecting most tissues are controlled by THs and GCs [14]. In this period, plasma GC levels rise markedly and coincide with those of circulating TH. These TH and GC variations are highly conserved in vertebrates and are considered to be an ancestral feature of all vertebrates [15]. In mammals, this period corresponds to the perinatal period when several biological processes including the maturation of the nervous system, the respiratory system, the immune system, the bones, and the intestine are induced. TH or GC deficiency results in abnormal development, i.e., severe mental retardation and growth defects or death. Furthermore, during this developmental period, GCs serve as a link between environmental inputs and physiological adaptation. However, in some circumstances, adaptation is associated with detrimental outcomes because this developmental transition period is particularly sensitive to environmental challenges in connection with the concept of developmental origin of health and diseases [16]. The interaction between TH and GC will then serve as an interface between environmental cues and TH-dependent developmental gene networks. Again, Xenopus is an ideal model because GC as mediators of emergency situations accelerate or delay metamorphosis to allow the animal to either escape a hostile environment or wait for more favorable conditions [17].

To further highlight the strength of our model, Xenopus metamorphosis is a post-embryonic transition occurring in free-living individuals; thus, it has no confounding effect from the mother’s TH/GC status. Moreover, the liver is a well-known target tissue of THs and GCs. The liver performs detoxification, production of serum proteins, endocrine functions, and metabolism of energetic substrates, which are modulated by THs or GCs [18,19]. Dysfunction of these two hormones also leads to abnormal physiology of this organ. During metamorphosis, biochemical changes include induction of the urea cycle (transition from ammonotelism to ureotelism), albumin synthesis, and globin switching, as well as changes in immune system function that impact the liver [20]. The liver is also the major site of proliferation of hematopoietic cells. GCs promote lipid storage, gluconeogenesis, and glycogenesis in larval amphibian liver, while THs promote glycolysis. The pattern of glycogen mobilization is consistent with the energy requirements of non-feeding climax stages. However, no specific consequences of TH and GC acting together are known.

In this work, we reveal the complexity of the crosstalk between these two hormones. We measured transcript levels of developing tadpole liver treated for 24 h with triiodothyronine (T_3_), corticosterone (CORT), or both hormones together using RNA-seq. The duration of the treatments for 24 h was chosen on the basis of our previous data showing that this timepoint provides desirable induction levels of gene transcription [21]. Our results highlight a crosstalk between the two hormonal pathways. Indeed, the effects of the cotreatment gave rise to transcriptional programs distinct from effects of the single-hormone treatments. Interactions comprised potentiated, antagonistic, and synergistic effects. Using systems biology tools, crosstalk genes that control cell proliferation were identified, and their expression profiles suggested a decrease in cell proliferation.

## 2. Results

### 2.1. Liver Differentially Expressed Genes following Thyroid Hormone and Corticosterone Treatment

To identify interactions between the TH and GC signaling pathways at the transcriptional level in the liver, tadpoles were treated for 24 h with T_3_, CORT, or T_3_ + CORT. Total RNAs were extracted from livers and subjected to deep sequencing. Raw data were analyzed with standard RNA-Seq processing procedures. Globally, a total of 3087 genes were differentially expressed (DE). More specifically, compared to the control, 2050, 25, and 1984 genes were significantly DE following treatment with T_3_, CORT, and T_3_ + CORT, respectively (Figure 1A, Appendix A). The induction fold (log_2_) varied from −8.88 to 8.37 for T_3_, from −0.55 to 2.6 for CORT, and from −7.2 to 8.38 for T_3_ + CORT. RT-qPCR carried out for 21 genes, to validate the RNA-Seq data, showed correlations between the RNA-Seq and RT-qPCR values (Figure 1B). Examples of the validation are shown in Figure 1C.

In order to establish the biological processes associated with the DE gene sets corresponding to each treatment, we carried out Gene Ontology analysis (Figure 1D). DE gene sets were analyzed whether they were up- or downregulated following treatment. Gene ontology analysis for the downregulated genes following the CORT treatment gave rise to terms (“GO terms”) involved in long-chain fatty-acid CoA synthesis and metabolism (Figure 1D, Appendix A). No significant enrichment was found for the upregulated gene set. T_3_ and T_3_ + CORT upregulated genes led to very similar GO terms mainly involved in Golgi and endoplasmic reticulum-mediated molecular transport, as well as protein localization (Figure 1D, Appendix A). T_3_ downregulated genes were particularly involved in lipid metabolism and signal transduction (Figure 1D, Appendix A). T_3_ + CORT downregulated genes were mainly involved in metabolism of organic cyclic compounds and small molecules (Figure 1D, Appendix A).

### 2.2. Characterization of Expression Profiles to Reveal Crosstalk between T_3_ and CORT

To address the extent of crosstalk between these two hormonal pathways, we first carried out expression profile-based clustering starting from the results of the DE genes. The aim of this procedure was to classify individual genes into different and specific “response types” or “clusters” based on significance differences in their expression levels among the four conditions (control, T_3_, CORT, and T_3_ + CORT) but independent of the magnitude of the gene expression levels. Thus, DE genes (3087 genes) were distributed in 26 clusters corresponding to 26 response types. Clustering analysis (see Methods) is presented in Figure 2. Each cluster is characterized by four letters. The first letters correspond to the change of expression compared to the control following T_3_, CORT, and T_3_ + CORT treatment, respectively (“d” for downregulated, “u” for upregulated, and “n” for no effect). The fourth letter was assigned to each cluster following the decision tree in Appendix A to indicate the effect the two hormones had on each other’s gene regulation for that response type.

We defined a response type as a case of crosstalk when the response to the hormone cotreatment is not explained by the simple addition of the responses to each hormone alone. For example, “dnd” and “unu” clusters correspond to response types where the genes are up- or downregulated by T_3_ and unresponsive to CORT. In contrast, an “und” cluster would indicate a crosstalk response type where the genes are induced by T_3_ and not CORT but repressed by cotreatment. In clusters 1 to 6, each hormone acts independently from the other, while clusters 7 to 26 correspond to crosstalk between the two signaling pathways. Different types of crosstalk are apparent, each corresponding to alternative regulation scenarios. The fourth letter for each cluster differentiates instances of no effect of one hormone to the other (“N”), potentiation of action (“P”), antagonism (“a”), reciprocal antagonism (“A”), or synergy (“s”) and reciprocal synergy (“S”). For example, clusters 1 and 2 (nddN and nuuN) show no evidence of crosstalk (no regulation with T_3_ and no difference between CORT and T_3_ + CORT).

Clusters 1 and 2, thus, correspond to genes regulated by CORT only. Similarly, clusters 3 and 4 (dndN and unuN) are instances of T_3_ regulation only. Clusters 5 and 6 (dddN and uuuN) correspond at most to the additive action of each of each hormone. In these clusters (1 to 6), there is no action of a hormone on the effect of the other; thus, they are granted the letter “N” for no effect (Figure 2, Appendix A). In clusters 12 (dnnA), 13 (dunA), 14 (udnA), 15 (unnA), and 25 (dnuA), the amplitude of the transcriptional response after cotreatment is diminished compared to the action of T_3_ and CORT alone, thus indicating reciprocal antagonism of action. A similar analysis concludes that clusters 7 (duna), 8 (nuna), 9 (dnna), and 16 (unna) correspond to an antagonism of CORT signaling by T_3_. In these clusters, the distance between CORT and T_3_ + CORT is superior to the threshold, and the distance between T_3_ and T_3_ + CORT is inferior (see Methods and Appendix A). Conversely, clusters 10 (ddna), 11 (dnda), 17 (unua), and 18 (uuna), correspond to an antagonism of T_3_ signaling by CORT. In clusters 19 (dddP) and 20 (uuuP), responses to cotreatments correspond to more than the sum of the effect of each hormone individually. This is also true with clusters 21 (nddS) and 22 (nuuS), with strong synergy of action. Strikingly, clusters 23 (nndS) and 24 (nnuS), which are not responsive to either T3 or CORT alone, are strongly regulated to both hormones, making them extreme cases of synergism of action. Cluster 26 (nnds) corresponds to an example of synergistic action in which only the CORT effect is superior to threshold (see methods).

We next sought to identify biological processes corresponding to the different types of responses (Figure 3).

First, the 26 clusters were sorted into seven categories on the basis of how genes respond to each hormone (Table 1, Appendix A). The gene ontology analysis was performed according to whether the genes were up- or downregulated.

Terms associated with the genes regulated only by T_3_ (category A) were protein transport and localization for the upregulated genes (Figure 3, Appendix A), and actin filament-based process and sodium transport for the downregulated genes (Figure 3, Appendix A). No term was found for the genes upregulated by CORT only (category B). Two genes were downregulated in response to CORT treatment and were associated with the Toll-like receptor signaling pathway and the regulation of the IFN-beta production. (Figure 3, Appendix A). In category C, genes equally downregulated by the two hormones belong to pathways involving metabolic processes and lipid metabolism (Figure 3, Appendix A), while those upregulated point to aminoacylation of tRNA and amino acids (Figure 3, Appendix A). Among the genes DE following T_3_ treatment, not DE by CORT but with their expression modified by the presence of CORT (category D), only upregulated genes gave rise to gene ontology terms: regulation of NLRP3 inflammasome complex assembly and positive regulation of IFN-beta production, both components of the innate immune system (Figure 3, Appendix A). Looking at DE genes following CORT and not T3 treatment but with their expression modified by the presence of T_3_ (category E), half are upregulated and associated with the GO term “rRNA metabolism” (Figure 3, Appendix A), and the others are downregulated and linked to the term “mitotic cell cycle” (Figure 3, Appendix A). The terms associated with upregulated genes following treatments with the two hormones (category F) are “innate immune response” and “stem cell and progenitor hematopoietic differentiation” (Figure 3, Appendix A), which reflect the larval to adult hematopoietic switch. The genes downregulated in this category are linked to terms such as DNA repair, DNA metabolism, and cell cycle (Figure 3, Appendix A). Lastly, genes DE only when the two hormones are used jointly (category G) gave rise to gene ontology terms only for those downregulated. They share the same processes with genes previously described: DNA repair, DNA metabolism, and cell cycle (Figure 3, Appendix A). Altogether, genes subject to crosstalk between T_3_ and CORT mainly induce the innate immune system and the hematopoietic function of the liver by acting on the stem cell and progenitor hematopoietic differentiation. Crosstalk slows down the cell cycle by inhibiting DNA repair and DNA metabolism, which are involved in the process of DNA replication.

### 2.3. Network Analysis of the Crosstalk between T_3_ and CORT

Network biology provides a way to integrate functional data into biological contexts and to communicate how treatments affect signaling and metabolic pathways and their interactions. Thus, we constructed a network of functional interactions by bringing together all the KEGG pathways containing at least one DE gene identified in our RNA-Seq analysis (Figure 4A). This allowed us to provide an integrated view of the functional interactions between gene products. The network encompasses 125 pathways out of 345 KEGG pathways in the database and contains 563 DE genes. The remaining DE genes were not integrated into the network analysis because they either failed to map to human orthologs, were not found in the KEGG database, or corresponded to unconnected nodes within the network.

The resulting network contained 3750 nodes (genes) and 15,391 edges (molecular interaction, reaction, and relation pathways representing systemic functions of the cell and the organism) (Figure 4B). Among the nodes in the connected component of the network, 563 genes (18% of all the DE gene) are DE genes regulated by T_3_ only (481 genes, 23,4% of this category), CORT only (one gene, 4% of this category), or by a crosstalk between the two hormones (82 genes, 4% of genes of this category).

Highly connected nodes constitute integration points that provide communication and signal propagation through networks. Such nodes with a connectivity of 20 or higher are hubs. They are key elements in the regulation of biological processes, and, because of their high connectivity, functionally perturbing them has a strong biological impact. The network here contains 402 hubs (10.72% of the nodes of the network). Among them, 69 are differentially expressed, 52 belong to the category regulated by T_3_ only (clusters 3 and 4), none belong to the category regulated by CORT only, and 17 are DE genes regulated by the crosstalk (*cdc7*, *pold3*, *socs1*, *ndufb10*, *hspa1a*, *msh6*, *skp2*, *e2f1*, *psmb6*, *myc*, *psme3*, *pmm2*, *eif2ak2*, *cdkn1b*, *tuba4a*, *apex1*, and *psmd6*) (Figure 4B). The hubs regulated by crosstalk belong to the clusters “dddP”, “uuuP”, “nndS”, “nnuS”, “dnuA”, “unnA”, “unua”, “dnda”, and “dnnA” with the number of edges (functional connections) ranging from 21 (*polD3*) to 72 (*myc*) (Figure 4C).

To further bring out the key molecular determinants of the crosstalk between T_3_ and CORT, the network was searched for “chains” of DE genes, i.e., groups of genes that are DE, as well as functionally connected. Two chains comprising 12 and four DE genes were found, namely, *myc*, *skp2*, *e2f1*, *cdkn1b*, *msh6*, *tuba4a*, *pmm2*, *apex1*, *pold3*, *tdg*, *hspa1a*, and *eif2ak2*, and *psmb6*, *psmd6*, *psme3*, and *casp6*, respectively. Among these 16 genes, 15 were regulated by crosstalk, and one (*eif2ak2*) was regulated by T_3_ only. These two chains contain a remarkable number of crosstalk-regulated hubs (14 out of the 17 hubs found in the network, Figure 4D). These groups of genes are, therefore, elements of the biological network specifically targeted by crosstalk regulation.

Next, we used gene ontology analysis to address the biological functions played by the 17 hubs specifically targeted by crosstalk. This analysis revealed that hubs are mainly associated with three terms: macromolecule metabolic processes, cytokine-mediated signaling pathway, and regulation of the G2/M transition of the cell cycle (Figure 4E, Appendix A).

We next looked for network features particularly targeted by crosstalk. We identified a subnetwork centered on two crosstalk regulated hubs, *psmb6* and *cdc7*. In order to help visualize the hierarchy of the first and second neighbors of *psmb6* and *cdc7*, we displayed first neighbors in a circle around each of them and second neighbors of *psmb6* and/or *cdc7* in an outer circle (Figure 4F). When we look at the first neighbors, *psmb6* and *cdc7* interact with few DE genes. Indeed, in this subnetwork, *psmb6* is in relation with three T_3_-regulated genes (blue dots: *psmd8*, *cdc25a*, and *sae1*) and two crosstalk-regulated genes (red dots: *psme3* and *psmd6*, all involved in the proteasome pathway). *cdc7* interacts with two T_3_-regulated genes (*sar1b* and *mcm5*) and no crosstalk-regulated genes. Interestingly, when looking at second neighbors, *psmb6* and *cdc7* are in connection with 60 T_3_-regulated genes and nine crosstalk-regulated genes (*myc*, *pmm2*, *dnajc1*, *aprt*, *casp6*, *skp2*, *usp8*, *hspa1a*, and *tuba4a*) (Figure 4F). This enrichment of DE genes is statistically significant (permutation test, z-score = 3.6, *p* < 10^−3^). *Orc1* is the only first neighbor in common with *psmb6* and *cdc7*. Thus, these two hubs regulated by crosstalk are part of a subnetwork, testifying to its important role in the hormonal regulation. We note that *psmb6* and *cdc7* are also involved in the G2/M transition of the cell cycle. We validated the results of the RNA-seq for *cdc7* and *psmb6* using RT-qPCR (Figure 4G).

## 3. Discussion

In this work, we provide the first transcriptomic analysis of the interaction between TH and GC treatments in the liver during development. Transcript-level measurements by RNA-seq identified genes regulated by hormonal crosstalk, and gene ontology and network analyses revealed the biological processes targeted by this crosstalk.

### 3.1. TH and GC Cotreatment versus Single-Hormone Treatments

Following cotreatment with T_3_ and CORT, most of the genes (2596; 84%) showed a profile corresponding to the response to T_3_ only. Among these genes were some involved in biological processes well known to take place in liver during metamorphosis. Specifically, upregulation of ornithine aminotransferase and ornithine decarboxylase 1 by T_3_ attested to the switch from ammonotelism to ureotelism in hepatocytes [22]. Furthermore, the switch from lipid storage to carbohydrate [23] was illustrated by genes associated with the GO term for regulation of lipid metabolic processes, which are downregulated following T_3_ treatment. The short duration of the hormonal treatments precluded detection of changes in genes associated with the larval to adult switch in serum protein production.

Conversely, very few genes displayed an altered expression profile following CORT treatment (27; 0.87%), suggesting that, in the absence of T_3_, liver remodeling is not transcriptionally sensitive to CORT or that CORT action is mediated by its nongenomic effects. The limited effect of CORT is not the result of the absence of GR or MR expression or the overexpression of an enzyme degrading CORT, as observed in our RNA-Seq dataset. With this limited set of DE genes, GO analysis only reported downregulation of genes involved in lipogenesis. 

Interestingly, a substantial number of DE genes (464; 15%) displayed expression after cotreatment distinct from the sum of the two single-hormone treatments, indicating interactions between these two hormonal signaling pathways. In addition, compared to the CORT treatment, the hormone cotreatment involved 17 times more genes differentially expressed associated with many significant GO terms, suggesting that crosstalk may have strong biological consequences. The different crosstalk effects are “potentiated” when there is a cooperative effect between the hormones, “antagonistic” when one or both hormones slow or reverse the action of the other, and “synergistic” when both hormones are needed to produce a transcriptional effect.

The different crosstalk response types (clusters) can be pooled into different groups depending on the differential expression following each hormonal treatment (Table 1). Thus, in Categories D and E, genes are DE after treatment with one hormone only, but, in the cotreatment, their expression is influenced by the presence of the other. In Category F, genes are DE following treatment with both hormones. A very well-established example is the gene *klf9* whose transcription is controlled by a “synergy module” present upstream of its promoter [11], although this is not the case in the liver where it is regulated by T_3_ only. In the G category, genes require the presence of both hormones to be DE and respond to the hormonal treatments.

On the basis of the literature, one might consider that variations of transcript abundance of *dio2* and *dio3* could underlie the crosstalk between T_3_ and CORT [9]. Indeed, CORT acts in enhancing the expression of the T_3_ activating enzyme *dio2* and in reducing the expression of *dio3* that inactivates T_3_. The resulting effect is to increase the tissue availability of active T_3_. This crosstalk mechanism does not fully occur here in the liver since *dio2* is not differentially expressed and *dio3* is downregulated to the same level either with T_3_ and CORT only or with the two hormones together. We can, therefore, wonder about the molecular mechanisms at the origin of the numerous expression profiles identified here. Direct binding of TRs or GRs to the DNA hormone-responsive element and interaction with the promoter of their respective response genes only account for a part of the observed variations. TRs interact with a limited number of transcription factors, but THs are responsible for variation in the expression of an important number of transcription factors and enzymes involved in chromatin organization/dynamic of transcription. GRs interact with other transcription factors [6] and can, thus, induce a variety of effects. Thus, TH and GC cotreatments result in complex chromatin remodeling and transcriptional status; however, further experiments are needed to provide additional insights into the molecular mechanisms underlying the hormonal crosstalk.

### 3.2. Role of TH and GC Crosstalk in the Control of Cell Division

Networks in systems biology provide the theoretical and technical framework to model functional interactions within the cell. This framework provides an objective description of the circuitry controlling the flow of matter and information. Key components of cellular activity correspond to hubs, which are control points and regulators. For example, network analysis as applied to cancer highlights hubs, in which mutations can highjack cellular activity. Hubs are often important among many cell types, explaining why anticancer drugs targeting hubs elicit numerous side-effects. In this work, we examined whether hormonal crosstalk is mediated by (or excluded from) hubs.

We found crosstalk between T_3_ and CORT involving several components of the cell cycle strongly pointing to regulation of cell proliferation. Thyroid hormones, through their nuclear receptors, can regulate the expression of many genes involved in control of the cell cycle [24]. However, because actions of THs are highly pleiotropic, their effects on proliferation are heterogeneous depending on the cell type, the cellular context, and the developmental status. TH receptors have been shown to play a tumor suppressor role, representing an opportunity to identify novel therapeutics in hepatocellular carcinomas [25]. GCs were also shown to regulate cell proliferation of different cell types, leading to G1 arrest [26] and to decreased DNA synthesis in hepatocytes [27]. GCs have been widely used as cotreatment for cancer patients [28] and the numerous studies aiming to understand their mode of action have succeeded in highlighting some of the actors. Cell-cycle components, such as *c-myc*, *p53, pRb*, *E2F*1, and *cyclin D1*, are some of the protagonists that we have also shown to be regulated in our studies [29]. In another study, *c-myc* was shown to be a hepatocyte antiproliferative agent [30].

Interestingly, in addition to the previous genes involved in G1 progression, T_3_ and CORT regulate key players at the G2/M transition of the cell cycle, e.g., by increasing the expression of 26S proteasome components [31]. One of them, *psmb6*, presents an interesting expression profile following T_3_ and CORT treatment. Indeed, each hormone used alone had no effect on *psmb6* RNA expression level, while the use of both hormones together led to an increase in its expression. The proteasome subunit beta type-6 is essential for the assembly of the 20S proteasome complex involved in protein degradation. Proteasome components are hubs as they are key regulators of many biological processes. One of the proteosome’s main targets during the cell cycle is cell division cycle 6 (*cdc6*) protein. This protein is required for loading of the mini-chromosome maintenance (MCM) protein complex onto the DNA at the origins of replication. After its essential role, CDC6 is phosphorylated and sent for 20S proteasome degradation, thereby avoiding a second round of DNA synthesis. Next, the cell division cycle 7 kinase (*cdc7*) phosphorylates MCM2 and MCM3, allowing the initiation of DNA replication in mitosis [32]. The transcript coding for *cdc7* is also regulated by T_3_ and CORT and subjected to crosstalk. Both hormones when used alone decrease *cdc7* RNA level, while their simultaneous presence leads to a decrease bigger than the sum of the reductions caused by the two single treatments. With less *cdc7*, the cell cycle is arrested at the G2/M checkpoint. Targeting *cdc7* was also considered to block cell proliferation of liver cancer cells [33]. The variation in expression levels of many other genes, including *mcm5*, *psme3*, *psmd6*, and *myc*, is in agreement with an antiproliferative effect of T_3_ and CORT crosstalk.

The physiological consequence of the inhibition of cell proliferation is unknown. On one hand, reducing the number of cell divisions can accelerate the metamorphic process, a phenomenon classically observed to escape unfavorable environment. In this context, CORT mediates the stress that allows the animals to respond and survive. On the other hand, going through the metamorphic process with fewer cell divisions can potentially lead to adverse effects. Thus, any advance in the understanding of stress-induced developmental plasticity and associated cost, as well as in the knowledge of the molecular mechanisms leading to crosstalk between T_3_ and CORT, will have strong fundamental and practical implications (medical, ecological, and agricultural, as well as on livestock production). 

## 4. Material and Methods

### 4.1. Animal Care and Hormonal Treatment of Tadpoles

*Xenopus tropicalis* adult frogs were obtained from our lab colony and maintained as previously described [34]. Following seminatural reproduction, tadpoles were raised until prometamorphosis as already described [34]. *Xenopus tropicalis* tadpoles at stage 47 were also obtained from the CNRS animal facilities (CRB-Rennes-France). Developmental stages were assessed using the Nieuwkoop and Faber (NF) normal table of development of *Xenopus laevis* (Daudin) [35]. Animal care was in accordance with institutional and national guidelines (University of Cincinnati IACUC animal use protocol number 06-10-03-01 and Cuvier Ethic Committee14845-2018042318127469v2). The 3,3′,5′-triiodothyronine (T_3_, T2752, Sigma, Saint-Quentin Fallavier, France) was added directly to the tank at a final concentration of 10 nM. Corticosterone (CORT, C2505, Sigma, Saint-Quentin Fallavier, France) dissolved in 100% DMSO (D8418, Sigma, Saint-Quentin Fallavier, France) was added to the tank to a final concentration of 100 nM [36]. All conditions received an equivalent amount of DMSO (0.001%). For treatment, 10 tadpoles at stage NF-54 were placed in a 1 L beaker containing 500 mL of dechlorinated tap water, where the hormones were added. Tadpoles were euthanized 24 h later by an overdose of anesthesia (0.01% MS222, Sigma, Saint-Quentin Fallavier, France) to dissect the liver [35]. Livers were snap-frozen in liquid nitrogen and stored at −80 °C. Three independent biological replicates were used for the RNA-Seq, and 10 independent replicates were used for RT-qPCR analysis.

### 4.2. RNA Isolation and Gene Expression Assessment

The RNA isolation was performed as previously described [21]. Total RNA was extracted from liver tissues with RNAble (Eurobio, Les Ulis, France) and further purified on spin columns (Qiagen miniKit, Les Ulis, France). RNA was quantified with a Nano-Drop ND-1000 UV/Vis spectrophotometer at 260 nm (Nano Drop Technologies, Wilmington, DE, USA). RNA integrity was measured via the Agilent 2100 Bioanalyzer with a minimum RNA integrated numerical value (RIN) of 7. Following DNAse treatment as indicated by the manufacturer (TURBODNAse, Ambion, Illkirch-Graffenstaden, France), reverse transcription of mRNA and quantitative PCR were carried out to quantify RNA abundance [21]. Primers were designed using Primer express (Applied Biosystems, Illkirch-Graffenstaden, France) (Appendix A). Raw results were processed using the 2^−ΔΔCt^ method. Data were normalized using the endogenous control *rpl8*. The endogenous control was selected according to NormFinder [37]. The results are presented as log_2_ fold-changes compared to the nontreated control. Statistical significance was addressed using a Mann–Whitney test.

### 4.3. RNA-Seq Data Processing

The FASTQC toolkit (http://www.bioinformatics.babraham.ac.uk/projects/fastqc/ (accessed on 23 March 2020)) was used to assess the sequencing run quality. Redundant reads were filtered, and the *fastx* toolkit (v 0.0.13) was used to clip the 3′ end of reads when the score dropped below 30 on the Sanger scale (Phred+33). Remaining reads were mapped on version 4.1 of the *X. tropicalis* genome [38] using bowtie 0.12.3 [39] with the following parameters: “*-5 10 -m1 -n2 -l28*”. Gene models used were an aggregation of gene models accessible in Ensembl (version 61) [40] and Xenbase [41]. Differential expression analysis was performed with DESeq [42] version 1.12 with the following parameters: *method =* “*pooled*”*; sharing-mode =* “*maximum*”*; fit-type =* “*parametric*”. Genes were considered statistically differentially expressed at an FDR ≤ 5%.

### 4.4. Clustering

Clustering of DE genes aimed to assign individual genes into specific “response types”. Since gene expression levels varied over five orders of magnitude (see MA-plots Figure 1A), expression values of each gene across the four treatment conditions (CTRL, T_3_, CORT, T_3_ + CORT) were transformed in order to model the dynamics of gene response independently of the expression level. To this end, read counts across the four treatment conditions were normalized by setting their average to 0 and their variance to 1 (Z transformation). Each treatment was then compared to CTRL, and the difference was used to determine whether the gene was upregulated (“u”), downregulated (“d”), or not regulated (“n”) after each treatment. The letters corresponding to the effect of each treatment (T_3_, CORT, T_3_ + CORT) designate the cluster names. For example, the cluster “dnd” corresponds to genes downregulated by T_3_, not regulated by CORT, and downregulated by T_3_ + CORT. It is important to note that the vertical axes shown in Figure 2 only represent normalized expression changes relative to the control. Absolute values displayed on these axes, including negative values, have no real meaning, and only differences relative to control are taken into account.

This clustering method is a gene-level transformation with no constraint on the actual number of genes within each cluster. This method contrasts with k-means clustering which tends to produce clusters with a similar number of genes and often misclassifies categories with a limited number of genes. It also has a better control of false negatives because it does not rely on multiple uses of a statistic with limited power, as is the case with differential analysis performed with few biological replicates (n = 3, as the current standard suggests) and many observables (~20 k genes). It is, therefore, expected that the clustering results deviate slightly from the DEseq analysis.

Compared to our previous work [42,43], we used an additional letter in the cluster name to provide a qualitative description of the regulation involved. This additional letter indicates whether the action of one hormone influenced the action of the other. By default (i.e., no crosstalk), the letter is “N”, while potentiated action (when one hormone potentiates the action of the other) is designated by “P”, antagonism of action (when one hormone antagonizes the action of the other) is designated by “a”, and mutual antagonism (both hormones mutually antagonize each other) is designated by “A”. Lastly, “s” and “S” indicate synergy, the difference between the two being that, with “S”, there is only a biological effect when the two hormones are present.

The decision tree (Appendix A) helped to define the fourth letter. It takes into account, firstly, the action of the hormones (d, u, or n) in each condition (T_3_, CORT, or T_3_ + CORT) and, secondly, if the distance between the effect of T_3_ or CORT and the effect of T_3_ + CORT is above a threshold (t) of 0.9, (i.e., if the effect of T_3_ + CORT is different from the effect of each hormone individually). When both distances were superior to threshold, the cluster was designed by “A” or “S”; when only one distance was superior to the threshold, the cluster was designated by “a” or “s”.

### 4.5. GO Analysis

The gene ontology (GO) analysis was performed using the GOrilla bioinformatics resources, with the human database as the background gene list, after mapping Xenopus gene IDs to their human orthologs.

### 4.6. Network Analysis

Human KEGG pathways (Kyoto Encyclopedia of genes and Genomes database) containing at least one DE gene were collected and merged to create a network as previously described [21], after mapping the Xenopus gene IDs to their human orthologs. Nodes with connectivity of 20 or more correspond to hubs. Cytoscape was used to analyze the network and to highlight the network properties [44]. Nodes correspond to gene products and edges represent functional connections between nodes. Hubs are defined as nodes with a degree of connectivity higher than 20 edges. Chains were identified by removing non-crosstalk genes and only keeping nodes with at least one connection.

## Figures and Tables

**Figure 1 ijms-23-13715-f001:**
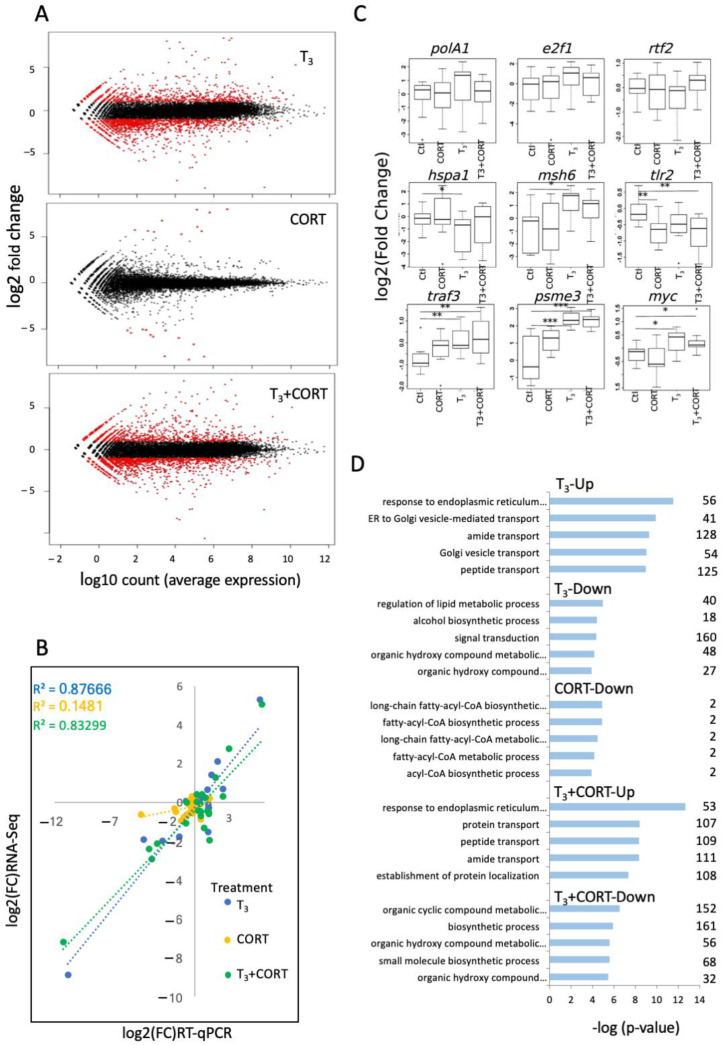
T_3_ and CORT treatments induce major changes in gene expression: (**A**) MA-plots for DE genes following each treatment; (**B**) validation of the RNA-Seq data by RT-qPCR carried out for 21 genes selected with an average expression level above 100 reads; (**C**) RT-qPCR validation of gene expression following T_3_, CORT, and T_3_ + CORT treatments (statistical significance according to Mann–Whitney test with * *p* < 0.05; ** *p* < 0.01; *** *p* < 0.001); (**D**) gene ontology analysis of the DE genes.

**Figure 2 ijms-23-13715-f002:**
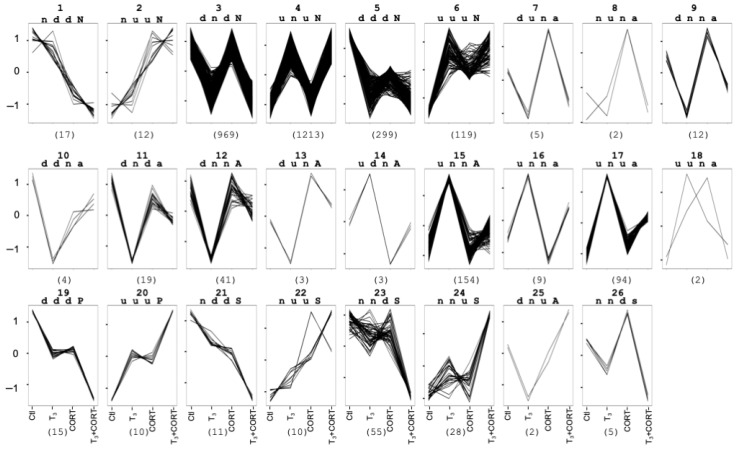
Expression profile-based clustering of the DE genes reveals crosstalk between T_3_ and CORT signaling. The first three letters describe the expression changes compared to the control. The fourth letter illustrates the type of crosstalk response: “N” for no crosstalk, “P” for potentiated, “A” for mutually antagonized, “a” for singly antagonized, and “s” or “S” for synergistic. With “s”, gene expression can be also regulated with each hormone individually, whereas, with “S”, gene regulation is strictly dependent on the action of both hormones simultaneously.

**Figure 3 ijms-23-13715-f003:**
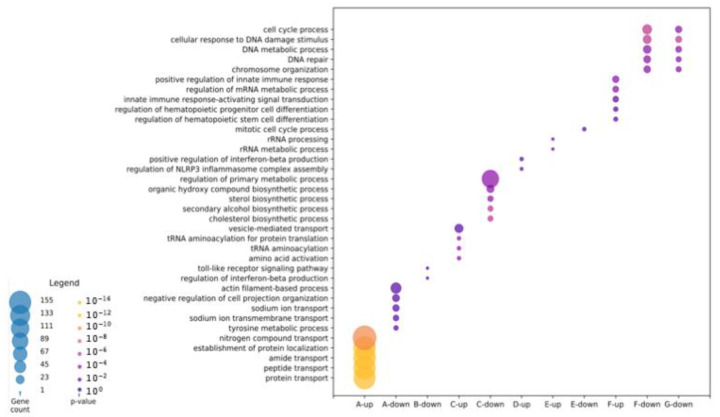
Gene ontology analysis highlights the different biological processes regulated by T_3_ and CORT and the crosstalk between the two hormones.

**Figure 4 ijms-23-13715-f004:**
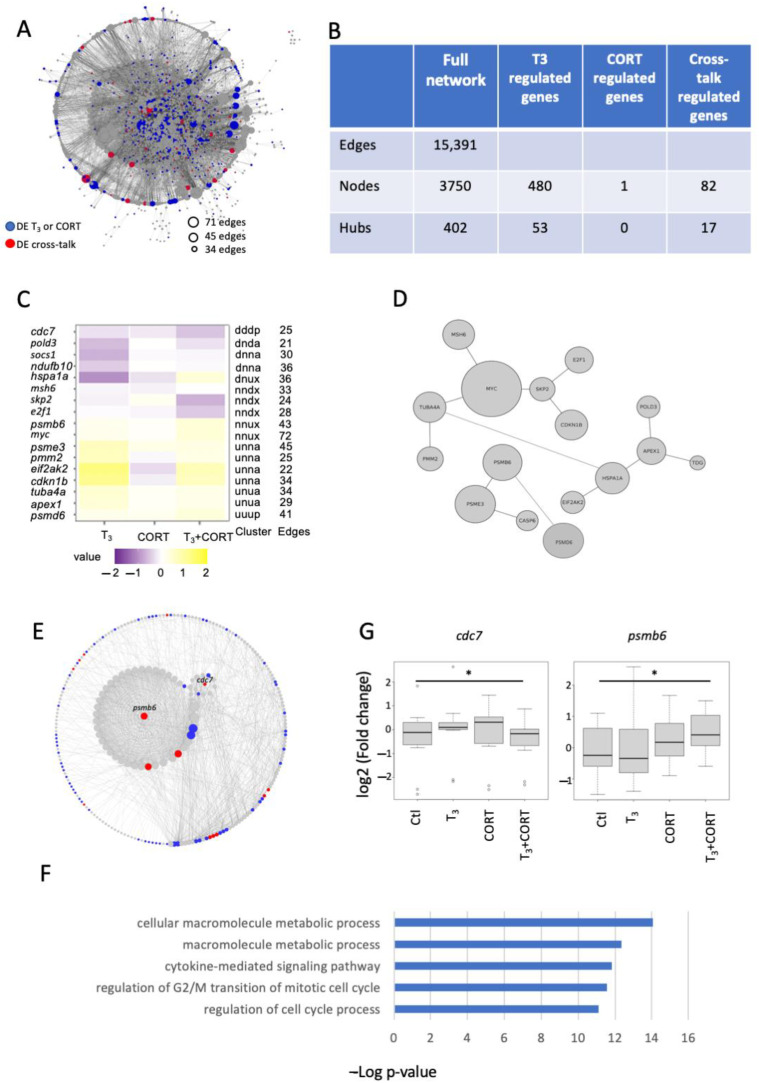
Network analysis of gene expression changes: (**A**) network for T_3_ and CORT DE genes (blue), crosstalk-regulated genes (red), and non-DE genes (grey). (**B**) Characteristics of the network. (**C**) Heatmap showing the expression pattern and the connections of the hubs. (**D**) Chains. (**E**) Gene ontology analysis of the crosstalk-regulated hubs compared to the network. (**F**) *psmb6*-*cdc7* subnetwork. (**G**) Validation of the crosstalk regulation of *psmb6* and *cdc7* by RT-qPCR. * *p* < 0.05 and >0.01.

**Table 1 ijms-23-13715-t001:** Transcriptional profiles of differentially expressed genes.

Category	Gene Number	% of DE Genes	Transcriptional Patterns—Clusters	Type of Gene Regulation
A	2180	70.5	dndN (3), unuN (4)	T_3_ only
B	27	0.87	nddN (1), nun (2)	CORT only
C	416	13.4	dddN (5), uuuN (6)	T_3_ and CORT independently of each other, no cross-talk
D	324	10.5	dnna (9), dnda (11), dnnA (12), unnA (15), unna (16), unua (17), dnuA (25)	Cross-talk: genes regulated by T_3_, not by CORT but influenced by CORT
E	20	0.64	nuna (8), nddS (21), nuuS (22)	Cross-talk: genes regulated by CORT, not by T_3_ but influenced by T_3_
F	35	1.2	ddna (10), dunA (13), udnA (14), dddP (19), uuuP (20)	Cross-talk: genes regulated by T3 and CORT with effects on the other hormone
G	85	2.75	nndS (23), nnuS (24), nnds (26)	Cross-talk: genes regulated only when the two hormones are present together

## Data Availability

The data presented in this study are available at SRA under the reference PRJNA883704.

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
