# Peer review of "Crosstalk between Thyroid Hormone and Corticosteroid Signaling Targets Cell Proliferation in *Xenopus tropicalis* Tadpole Liver"

_ijms, 2022, doi:10.3390/ijms232213715_

Round 1
Reviewer 1 Report
Comments for the authors:
In this study, the authors conducted a transcriptional analysis of the genes regulated by thyroid hormones (TH) and glucocorticoids (GC) in the liver of Xenopus tropicalis tadpoles to investigate the crosstalk between TH and GC pathways. Furthermore, the authors used gene ontology and biological network analyses of changes in gene expression to identify a sub-network centered on two genes regulated by the TH–GC crosstalk, psmb6 and cdc7, which regulate the G2/M transition.
The methodology is sound, and the manuscript is well written. The data presented in this manuscript would contribute to the progression of our understanding regarding the crosstalk between TH and GC not only in the liver but also in other organs.
The authors should consider one minor comment; however, the reviewer recommends the publication of this manuscript.
Comment:
1. The authors should explain the rationale for the concentrations used for T3 and CORT. Why did the authors use 10 nM TH and 100 nM CORT for the hormonal treatment of tadpoles?
Author Response
Reply to reviewer 1
1)
We used 10 nM TH and 100 nM CORT for the hormonal treatment of tadpoles because these concentrations were published earlier and correspond to the concentrations showing a synergy in the tail regression.
I add this reference in the methods section.
Ronald M Bonett 1 , Eric D Hoopfer, Robert J Denver. Molecular mechanisms of corticosteroid synergy with thyroid hormone during tadpole metamorphosis. Gen Comp Endocrinol. 2010 Sep 1;168(2):209-19. doi: 10.1016/j.ygcen.2010.03.014.
Reviewer 2 Report
In the current study, the authors investigated differentially expressed genes in the livers of Xenopus tropicalis tadpoles exposed to TH, GC, or both. The experiments are well-designed and the data are presented with high quality. The results suggest TH is associated with GC, regulating cell cycle in the liver. The study extends the knowledge of the interactions between TH and GC in development. I have a few minor suggestions below:
1. Could the authors add a venn diagram in Figure to show relationships of DEGs among 3 groups?
2. Figure 1C, what was the reason for choosing those genes as examples? Are these the only genes that " with the average expression level above 100 reads"?
3. I was confused by Figure 2. 1) What does the amplitude represent? 2) Does"Cl" mean "control"? If yes, please keep consistency with Figure 1, use either "CTL" or "Cl" but not both. Then why does genes expression in the control group also show up and down? Whom they were comparing with?
4. Honestly, Figure 2 does not make much sense to me. The logic between Figure 1 and 3 are more close. Is it possible to move Figure 2 to supplementary data?
5. Line 166, the first sentence is not very clear. Which clusters do "these" point to? Line 175,176, what do "superior" and "inferior" mean?
6. Please proofread the manuscript more carefully, the fonts of methods and some reference marks are inconsistent.
Author Response
Reply to reviewer 2
1)
Venn diagrams are popular representations of overlaps between sets of items. Not surprisingly, they are very commonly used in functional genomics to provide a graphical summary of differential analyses.
Unfortunately, despite providing an accurate description of the raw output of differential analysis softwares, their biological interpretation is much more cumbersome. The problem stems from the fact that the different sectors of Venn diagrams do NOT represent sub-datasets defined on equivalent statistical parameters. The nature of the argument is probabilistic and is as follows.
Functional genomics (almost) always relies on a very limited number of biological replicates, typically three and rarely more. Single biological replicates are not-uncommon either. In term of probabilistic modeling of variance (a.k.a. ANOVA-like differential analysis), statistical tests are therefore constrained to a very limited power, and the number of false negatives is expected to be correspondingly high.
In this context, overlapping sectors (i.e. sectors corresponding to genes DE in two conditions) do suffer from an enhanced rate of false negative, thereby strongly underestimating the number of genes corresponding to this sector. For example, if the rate of true positives in two sectors is 0.75, then the rate of true positives in the overlapping sector is 0.75 x 0.75 = 0.5625; almost a flip coin experiment. This bias is even stronger with sectors overlapping with more than two areas (and so on). In virtually all published Venn diagrams of DE genes, the inner sectors have very low gene counts, which clearly results from this fact. In other words, the rate of false negative in each sector is not the same, which prevent an accurate description of biological responses (although it correctly depicts the output of differential analysis).
Unfortunately, there are very little ways around this, and even non-parametric are not better alternatives because of their intrinsic low statistical power and their need of even more biological replicates. The popular DESeq differential analysis software uses statistical 'tricks' to artificially increase the number of biological replicates to enhance statistical power, but this is still limited.
In the present work (and our previous work before: Buisine et al, Cells 2021), we decided to depart from Venn diagrams and use instead a clustering strategy relying less on p-values. In short, and as detailed in the METHODS section, we use all genes which are DE in at least one experimental condition, and we derive profiles based on their normalized expression level. The key intuition here is that all DE genes, in any experimental condition, are subject to the same statistical power and does not suffer from the biases imposed by Venn diagrams. We agree that Figure 2 is complex, and this certainly reflects biological complexity, but this allows us to classify gene responses onto a number of response types (cross-talks, T3 only etc). In fact, the resolution of the gene ontology analysis shown Figure 3 directly results from the clustering analysis Figure 2.
It is clear that in addition to being inaccurate (and probabilistically unsound), replacing Figure 2 with a number of Venn diagrams will fail to reach the resolution and sensitivity needed for highlighting key biological processes in gene ontology.
Our feeling is that Figure 2 is an essential component of our analytical strategy.
2)
To validate the RNA-Seq by RT-qPCR, we chose 21 genes with the average expression level above 100 reads. Genes shown in Figure 1C are just examples to illustrate the validation of the RNA-Seq.
3)
- The amplitude refers to the fold change relative to the control. We completed to legend of the graph to make this more explicit.
- Yes Cl means control. We made the suggested changes, and modified Figure 2 so that it is more consistent with Figure 1.
- Good point. The fact that control group shows ups and downs is fully expected. The point here is to compare changes of gene expression relative to the control and the need to normalize data (with the very standard Z transformation -see METHODS section-) as we did. Accordingly, when gene expression goes up and down relative to the control after two treatments, this shows up as if the control group goes up and down. These graphs only display changes of gene expression relative to the control group. Therefore, the fact that control groups goes up only means that gene expression in treated groups goes down. The number of the vertical axis is not an expression level, but rather a normalized expression level, relative to the entire cluster. Normalized expression values are computed with the following constrain (Z normalization): for each gene of the cluster, the sum of all expression levels is set to zero. Therefore, if T3, CORT and T3+CORT all go down relative to the control by a (normalized) value of -0.3, then, given that CTL + T3 + CORT + T3+CORT = 0, it follows that CTL = +0.9. Conversely, if T3, CORT and T3+CORT all go up relative to the control by a (normalized) value of +0.3, then CTL = -0.9.
This comment highlights that normalized data might be hard to understand. We therefore update the METHODS section to make it clearer:
It is important to note that the vertical axes shown Figure 2 only represent normalized expression changes relative to the CTL. Absolute values displayed on these axes, including negative values, have no real meaning and only differences relative to CTL are taken into account.
4)
As explained above, Figure 3 is a natural extension of Figure 2, and we would find counter-intuitive to displace it to supplementary material. Again, Figure 2 displays essential results on which Figure 3 is based. The logic between Figure 1 and 3 is not the same. As stated in the manuscript, Figure 1 corresponds to the differential analysis, and depicts functional categories affected with each treatment. Importantly, taken as such, it does not discriminate between crosstalk and non-crosstalk response genes, which is the heart of the biological question we ask. In contrast, Figure 2 regroups genes based on their responses and discriminates crosstalk vs non-crosstalk genes. Figure 3 then highlights the biological processes specifically affected by crosstalk genes (which couldn't be addressed Figure 1). Therefore, Figure 2 is not accessory; it is rather a key part of our work. We feel that moving Figure 2 in supplementary materials would weaken the strength of our work as it would reduce its strategic importance.
5)
Line166: “these” are for clusters 1 and 2 (nddN and nuuN). In these clusters genes are not regulated by T3 and there is no difference between CORT and T3+CORT. In these clusters genes are regulated by CORT only. We up-dated the manuscript accordingly.
Line 175 and 176 “superior” and “inferior” refer respectively to the difference in the normalized expression level between CORT and T3+CORT, and T3 and T3+CORT. In these clusters (7 duna, 8 nuna, 9 dnna, 16 unna) the difference between CORT and T3+CORT is bigger than the difference between T3 and T3+CORT and these differences are respectively superior and inferior to the threshold. More mathematically, we looked at whether | T3 – CTL | £ THRESHOLD, | CORT – CTL | £ THRESHOLD... with an empirically defined threshold (see METHODS). Therefore, in these clusters T3 antagonizes the effects of CORT.